# A Review of Acquired Autoimmune Blistering Diseases in Inherited Epidermolysis Bullosa: Implications for the Future of Gene Therapy

**DOI:** 10.3390/antib10020019

**Published:** 2021-05-17

**Authors:** Payal M. Patel, Virginia A. Jones, Christy T. Behnam, Giovanni Di Zenzo, Kyle T. Amber

**Affiliations:** 1Department of Dermatology, University of Illinois at Chicago College of Medicine, Chicago, IL 60612, USA; ppate289@uic.edu (P.M.P.); valvara2@uic.edu (V.A.J.); calbaz2@uic.edu (C.T.B.); 2Rush University Medical Center, Division of Dermatology, Department of Otorhinolaryngology—Head and Neck Surgery, Rush University, Chicago, IL 60612, USA; 3Laboratory of Molecular and Cell Biology, IDI-IRCCS, 00167 Rome, Italy; G.DiZenzo@idi.it; 4Rush University Medical Center, Department of Internal Medicine, Rush University, Chicago, IL 60612, USA

**Keywords:** gene therapy, epidermolysis bullosa, autoimmunity, autoimmune blistering disorder, collagen XVII

## Abstract

Gene therapy serves as a promising therapy in the pipeline for treatment of epidermolysis bullosa (EB). However, with great promise, the risk of autoimmunity must be considered. While EB is a group of inherited blistering disorders caused by mutations in various skin proteins, autoimmune blistering diseases (AIBD) have a similar clinical phenotype and are caused by autoantibodies targeting skin antigens. Often, AIBD and EB have the same protein targeted through antibody or mutation, respectively. Moreover, EB patients are also reported to carry anti-skin antibodies of questionable pathogenicity. It has been speculated that activation of autoimmunity is both a consequence and cause of further skin deterioration in EB due to a state of chronic inflammation. Herein, we review the factors that facilitate the initiation of autoimmune and inflammatory responses to help understand the pathogenesis and therapeutic implications of the overlap between EB and AIBD. These may also help explain whether corrections of highly immunogenic portions of protein through gene therapy confers a greater risk towards developing AIBD.

## 1. Introduction

Autoimmune blistering diseases (AIBD) are rare diseases with significant morbidity and mortality [1,2]. AIBD are caused by autoantibodies targeting various skin antigens. In contrast, epidermolysis bullosa (EB) is a group of inherited blistering disorders caused by mutations in various skin proteins [3]. AIBD and EB often have the same protein targeted through antibody or mutation, respectively (Table 1).

EB encompasses a group of inherited skin fragility diseases marked by blisters which may erode and lead to ulcers in the skin and mucous membranes [4]. Recently, EB has been grouped into four expansive categories based on the location of tissue separation within the basement membrane zone (BMZ) [4]. These categories include the simplex forms (EBS), the junctional forms (JEB), the dystrophic forms (DEB), and the newest subtype, known as the Kindler syndrome [4]. In this review, we will focus on the JEB and DEB forms which involve tissue separation, namely within the lamina lucida, within and below the lamina densa confined to the upper papillary dermis, respectively [4].

While EB and AIBD are distinct entities, there are reports of spontaneous development of AIBD in patients with pre-existing EB. Likewise, several groups have noted the prevalence of anti-skin antibodies in EB patients. Although the pathogenicity of these antibodies remains elusive, some speculate that chronic inflammation in EB combined with antigen unmasking can lead to a break in immune tolerance, resulting in development of AIBD.

Gene therapy serves as a promising therapy in the pipeline for treatment of EB. Through various vectors, different technologies lead to correction or expression of functional portions of collagen [3,5]. However, with great promise, the risk of autoimmunity must be considered. For example, corrections of highly immunogenic portions of protein may confer a greater risk towards developing AIBD. For illustration, Figure 1 details the location of mutations in collagen VII and collagen XVII with respect to the immunogenic epitopes in AIBD. As such, we reviewed the literature concerning the development of AIBD and anti-skin in EB patients.

## 2. Presence of Anti-Skin Antibodies in EB

Although EB is an inherited disorder caused by genetic defects, autoantibodies to skin antigens have been demonstrated in the sera of patients with EB [14,15,16,17]. Overall, patients with EB have significantly higher antibody titers against components of molecules responsible for cell adhesion, such as desmoglein 1, desmoglein 3, collagen XVII, BP230, and collagen VII compared to controls (Table 2) [7]. Similarly, 40.6–90.6% of patients with EB have significant antibody titers against collagen type III, IV, and V, as well as laminin [14]. Different subtypes of EB have varying prevalence of these circulating autoantibodies [14,15,16]. For instance, patients with the generalized forms of EB had significantly higher antibody titers (2–5-fold) against desmoglein 1, desmoglein 3, collagen XVII, BP230, and collagen VII compared to patients with other EB forms [15,16]. This difference in antibodies concentrations was accentuated when the sera of patients with recessive DEB (the most severe EB subtype clinically) was compared to that of patients with EBS, as recessive DEB patients had 7–11-fold higher antibody titers against collagen XVII, BP230, and collagen VII [15]. Furthermore, a direct correlation was found between disease severity (Birmingham EB Severity scores) and the concentrations of the antibody titers against collagen XVII, BP230, and collagen VII [15,16].

The implications of elevation in serum concentrations of autoantibodies in EB are not yet known, but autoantibodies against collagen VII have been shown to induce blistering in humans and experimental models [18]. IgG4 dominated the autoimmune response in patients with collagen-VII-specific antibodies [19]. High levels of IgG4 autoantibodies and other circulating anti-collagen VII autoantibodies have been detected in a majority of recessive DEB patients, independent of the *COL7A1* mutation type or quantitative collagen VII levels [16,20,21]. In epidermolysis bullosa acquisita (EBA), autoantibody-induced tissue damage against collagen VII contributes to blistering [22]. Complement activation through both the classical and alternative pathways have been implicated in disease pathogenesis but the alternative pathway appears to be predominant [23,24]. Autoreactive IgG and immune complex-FcγR binding initiate an inflammatory complement cascade resulting in extravasation of neutrophils, release of proteolytic enzymes, and reactive oxygen species [25]. Moreover, T cells may perpetuate tissue damage in EBA through association with immune complexes and neutrophils [26].

Anti-collagen XVII antibodies trigger subepidermal blistering in this bullous pemphigoid (BP) model via complement activation and non-complement patterns [27,28]. The IgG4 subtype was found to induce inflammation by activating leukocytes in a non-complement fixing pattern or by binding collagen XVII in a Fc-independent manner causing dermo–epidermal junction (DEJ) separation in BP [29]. Elevated anti-collagen XVII autoantibodies are even linked to more active and severe disease, as well as poorer prognosis [30,31,32]. However, negative direct and indirect immunofluorescence test results in most cases suggest that circulating autoantibodies are not pathogenic [33].

Moreover, immune-mediated complications and disease pathology have been described in EB patients, including celiac disease, amyloidosis, post-infectious glomerulonephritis, and IgA nephropathy [34,35,36,37]. Among cutaneous disease, a few reported cases of autoantibodies causing concurrent AIBD in patients with inherited EB include: EBA in a patient with dominant DEB [20], EBA in a patient with recessive DEB [38], and BP in a patient with JEB [39] (Table 3). In each of these cases, the acquired AIBD was resistant to common therapy, patients had minimal clinical improvement, and one patient with dominant DEB even died from severe hypoalbuminemia and anemia [20]. The authors speculate that genetic modifiers or environmental factors may help explain why some patients with positive serology exhibit clinical disease while others do not. Although not fully understood, the authors propose that chronic blistering and inflammation due to altered protein synthesis and structure in EB [16,38] contribute to the immunologic recognition of “self.” Alternatively, it is plausible that activation of autoimmunity is both a consequence and cause of further skin deterioration in EB due to a state of chronic inflammation. Herein, we review the factors that facilitate the initiation of autoimmune and inflammatory responses to help understand the pathogenesis and therapeutic implications of the overlap between EB and AIBD.

## 3. Altered ECM Properties

In DEB, dysfunctional collagen VII induces inflammation even with minor injury or trauma [40]. This structural compromise coupled with increased tumor growth factor-β (TGFβ) and its downstream effects on fibroblast differentiation promote the biosynthesis and remodeling of the extracellular matrix (ECM) [40]. Primary fibroblasts found in DEB skin demonstrate a pro-fibrotic profile, along with increased levels of context-dependent fibroblast activator, WNT-5A [41], thrombospondin-1 [42], collagen V, collagen XII, and integrin α3β1. Additionally, loss of collagen VII decreases the levels of basement membrane-associated proteins and increases proteins linked to the dermal matrix, TGF-β, and metalloproteases [43,44]. Altogether, the chronicity of damage and increased dermal stiffness promote inflammation and scarring within the DEB dermis [45,46].

The mutant protein inherited in EB may be sensitive to such inflammation and protease cleavage, possibly resulting in the generation of neo-epitopes [47]. An immunocompromised district is a vulnerable area of skin resulting from an inherited or acquired process that causes dysregulation of the local inflammatory response [48]. It is likely that repeat mechanical injury in EB compromises the DEJ, thus increasing antigen presentation and subsequent epitope spreading, resulting in the activation of autoimmunity in EB [49]. Likewise, epithelial barrier disruption through triggering factors, such as surgical trauma, ultraviolet radiation, infections, and necrosis have been associated with development of AIBD [50,51,52,53,54].

Flightless I (Flii), an actin remodeling protein and nuclear receptor co-activator has been implicated in cell adhesion and intracellular signaling [55,56]. Disruptions in levels of Flii affect wound healing through effects on TGFβ signaling and ECM reorganization [56]. As such, levels of Flii are increased in blistered skin of EB patients, and Flii over-expression in experimental EBA decreased expression of proteins that make up cell–cell tight junctions [57]. The interaction of Flii with tight junctions may impair formation of tight junction protein complexes and function of the epidermal barrier [57]. Moreover, topical application of Flii-neutralizing antibodies improved healing in experimental EBA [56]. Additionally, increased levels of Flii have been shown to result in increased pro-inflammatory cytokine production in a T helper (Th)2 cells pattern in atopic dermatitis, and poor mucosal healing in ulcerative colitis [58,59]. Thus, altered levels of Flii in EB and EBA may contribute to the delayed healing and skin fragility [57].

## 4. Eosinophilia and Elevated IgE

Conventionally, EB is considered a paucicellular blistering disorder but individual reports of an inflammatory infiltrate in EB patients can be found [60]. The most commonly encountered infiltrate is eosinophilic, but leukocytes, lymphocytes, and neutrophils have all been documented in various types of EB [61,62,63,64,65,66,67]. Notably, three patients with JEB and extensive eosinophilic infiltration were reported with a common mutation in exon 5 of *COL17A1* [64]. Eosinophilia and eosinophilic infiltration are prototypic features of BP [50,51] where eosinophils play a crucial role in disease pathogenesis [68]. Therefore, the authors speculate if an autoantibody against truncated *COL17A1* evoked an immune reaction involving eosinophils or whether a common immunologic mechanism exists between BP and JEB [64].

Eosinophils are deemed vital to anti-collagen XVII IgE-mediated and FcεRI-dependent DEJ separation in BP [69,70]. IgE are important in presenting allergens to Th2, which produce IL-4 and IL-5 [71]. IL-5 also plays an important role in activating eosinophils that promote BP pathogenesis [72]. Degranulation of eosinophils releases several toxic proteins, including eosinophil cationic protein, which can induce keratinocyte production of IL-5, generating a positive feedback loop which may perpetuate BP [73]. Furthermore, eosinophils also act as a mediator of pruritus by exerting influence on peripheral nerves and the autonomic nervous system, as well as through local production of IL-31 [68,74,75]. Therefore, it is plausible that in EB cases where eosinophilic infiltration is accompanied by elevations in IgE or Th2 cytokines, the disease course may worsen likened to that of BP.

Moreover, another case of a DEB patient with features of both eosinophilia and elevated IgE has also been observed [61]. Elevations in IgE are seen in BP and correlated to disease severity and pathogenesis [69,70]. A humanized mouse model expressing a truncated form of collagen XVII, through dysfunctional NC16a, resulted in spontaneous BP-like inflammation characterized by severe itch and a defective skin barrier, along with hyper IgE and immune cell infiltration [76]. However, IgE or components of adaptive immunity were not linked to cause severe itch in these mice. Similar to BP, elevated levels of anti-collagen VII IgE autoantibodies were found in the sera of EBA patients and correlated to the inflammatory phenotype [77]. IgE specific to collagen XVII and BP230 were associated with inflammatory cells (eosinophils and mast cells) [78], suggesting that the anti-collagen VII IgE are of pathogenic relevance, and thus may exacerbate EB severity.

## 5. Pro-Inflammatory Microenvironment Due to Cytokine Dysregulation

High levels of pro-inflammatory cytokines, such as IL-6 [79,80], interferon-γ (IFN-γ), IL-1β, IL-2, and tumor necrosis factor- β (TNF-β) are present in patients with inherited EB, with the greatest increase in patients with recessive DEB [15,17]. Elevations in the circulating levels of IL-6 were also related to the severity of the disease and with anti-skin antibody levels [17]. IL-6 is known to promote the production of acute-phase proteins and Th17 proliferation, previously found to induce a shift from innate to acquired immunity [81,82]. Moreover, elevation in IL-6 levels was also found in the sera and blister fluid in EBA and BP cases [83]. The role of IL-6 in BP pathogenesis can be attributed to its pro-inflammatory drive and indirect activation of matrix metalloproteinase-9 (MMP-9), shown to weaken hemidesmosomes in BP [84]. Similarly, MMP-9 and IL-8 are not only upregulated in circulation, but also in the blister fluid of EBS, JEB, and DEB patients [85,86]. Both MMP-9 and IL-8 have been reported in other AIBD, such as BP and pemphigus vulgaris but their pathologic significance in EB and overlapping AIBD needs elucidation [68,87,88].

Pro-inflammatory chemokines, such as chemokine C-X-C motif ligand 12 (CXCL12) and high mobility group box 1 (HMGB1) [89,90] are increased in EB patients [91], sometimes in correlation with the affected body surface area [91]. CXCL12 and HMGB1 are both involved in the stress-induced recruitment of stem cells to damaged tissue and bone marrow mesenchymal stem cell (MSC) migration into wound cells, suggesting that together they promote wound healing in EB [92,93,94,95]. Serum levels of chemokine (C-C motif) ligand 21 (CCL21) are lower in EB patients but tissue samples show increased CCL21 expression [91]. Since CCL21 has been shown to influence MSC migration into the skin, the gradient between serum and tissue levels of CCL21 could promote MSC migration into tissues [96].

In a humanized mouse model expressing truncated collagen XVII, increased production of IL-1β was found to trigger keratinocytes to produce thymic stromal lymphopoietin, held responsible for initiating inflammation or itch, a common symptom in both BP and JEB [76]. Likewise, IFN-γ, another important pro-inflammatory marker, is expressed in high quantity in both BP blister fluid and EBS [97,98,99]. IFN-γ-mediated release of cytokines propagates a cell-mediated immune response that contributes to the autoantibody-induced blister formation in BP, along with IL-1β [98]. In addition, IFN-γ was also shown to positively correlate with anti-skin antibodies in EB, suggesting a common link between cytokine milieu and disease pathogenesis in EB and AIBD [100].

In contrast to IFN-γ, a negative correlation was found between IL-5 and autoantibodies to collagen VII and BP230 [17]. IL-5, along other cytokines, also plays an important role in eosinophil activation, survival, chemotaxis, and degranulation [68,101,102]. Moreover, IL-5 activated eosinophils induce subepidermal blistering at the DEJ in BP [72], further supporting the finding that IL-5 elevation is correlated to severity of BP [103]. Perhaps, local IL-5 levels combined with antigen unmasking drive autoimmunity to EBA and BP. Alternatively, IL-5 is previously shown to promote induction of CD4^+^CD25^+^ T-regulatory cells that are involved in the suppression of autoimmunity [104].

## 6. Dysregulated Inflammatory Response and Blister Formation

When subjected to trauma, keratinocytes display increased sensitivity to autoantibodies [105]. Collagen XVII is a known inhibitor of keratinocyte migration, while its shed ectodomain leads to stabilization and cell immobilization [67]. In nonlethal JEB, the absence of collagen XVII or dysfunctional interaction between laminin-332 and collagen XVII is speculated to promote keratinocyte migration [67,106,107]. Thus, the weakened attachment of keratinocytes to the basement membrane, increased keratinocyte sensitivity to circulating autoantibodies, and subsequent expression of eosinophil chemotactic factors enhance deposition of antibodies and facilitate blister formation [108].

However, blister formation has a much more complex etiology where the interplay between cytokines, chemokines, and MMP is important. In patients with JEB, anti-collagen XVII autoantibodies trigger the release of inflammatory cytokines that may exacerbate DEJ separation [109]. In vitro, when JEB-derived (collagen-XVII-deficient) epidermal keratinocytes are exposed to inflammatory stimuli (ultraviolet B radiation, lipopolysaccharide, phorbol 12-myristate 13-acetate, and tumor necrosis factor), an abnormally high IL-8 response is seen [109]. As a chemotactic agent, IL-8 contributes to neutrophil recruitment [110]. In turn, re-induction of collagen XVII expression normalizes this response (against lipopolysaccharide and ultraviolet B radiation), suggesting that it may serve as a pathway for inflammation and subsequent lesion formation in the skin of collagen-XVII-deficient EB patients [109]. Although not fully elucidated, antibody-mediated disruption of interactions between collagen XVII and other components of the BM may result in a pro-inflammatory response.

In the complement-independent pathway, the binding of autoantibody to collagen XVII and the subsequent internalization of these immune complexes causes a depletion of collagen XVII from cell surface [111]. This results in the formation of collagen XVII-deficient hemidesmosomes that weaken the adhesion in a patient’s skin [111]. Thus, BP-IgG may induce or exacerbate skin fragility by itself [28].

## 7. Bacterial Infections

An additional hypothesis for autoimmunity is related to molecular mimicry, where exposure to a foreign pathogen with a similar sequence to self-antigens incites autoimmunity through the activation of T and B cells [112]. Typically, EB patients are prone to bacterial colonization and wound infections [113,114]. Alteration of microbial diversity may be one of the inciting factors for the activation of autoimmunity in EB patients [15]. Repeat bacterial infections in the presence of chronic skin fragility and mechanical insults in DEB patients, promote a cycle of inflammation that results in an altered ECM architecture, dermal stiffening, and development of a mutagenic environment [15,115,116,117]. Therefore, molecular mimicry is likely to occur in EB.

Dysfunctional collagen VII affects the ability of lymphoid conduits to successfully sequester cochlin within secondary lymphoid organs, resulting in a loss of the innate immune response to bacterial challenges [116]. This offers strong evidence that the increased susceptibility to bacteria in recessive DEB and the aforementioned cytokine imbalance induce a chronic inflammatory response that could activate autoimmunity and/or worsen basal EB lesions. Further strengthening the importance of alteration of microbial diversity in BP is a study that reports clinical improvement of BP with antibiotics [118].

Taken together, these studies establish that autoimmunity and inflammatory responses are frequently activated in EB, potentially setting the stage for the formation of other autoimmune diseases.

## 8. Implications of Autoantibodies in Gene Therapy

Recent advancements in the understanding of EB pathogenesis have allowed researchers to identify novel treatment options, including gene therapy. Initial success in gene therapy was uncovered for JEB patients with a LAMB3 mutation, who received genetically engineered epidermal sheet grafts overexpressing an ex vivo, retroviral full-length LAMB3 transgenic product [119,120]. LAMB3 expression was maintained within the holoclonal epidermal stem cells and laminin 332 was found in the DEJ until 21 months [120]. However, gene therapy-mediated expression of a functional protein runs the risk of inducing autoimmunity. Fortunately, these study patients did not generate an immune reaction to the antigenic laminin β3 chain [121], likely because the selected patients had missense mutations involving a single amino acid or small deletions [119,120]. On the other hand, JEB patients with null mutations and fatal disease would be expected to develop immunoreactivity, and laminin-332 expression in skin via gene therapy would not correct severe upper respiratory, kidney, or internal disease [3]. Notably, autoantibodies against laminin β3 are also uncommon in patients with laminin-332 pemphigoid, occurring in less than 30% of patients [122]. However, autoantibodies against laminin α3 are present in close to 90%. This suggests that LAMB3 may be a less immunogenic target, thus contributing to the success of LAMB3 correction.

Gene therapy in recessive DEB patients is even more challenging due to the large size of *COL7A1* and the increased immunogenicity of NC1 [123]. Epidermal sheet grafting maintained collagen VII expression in the primary collagen-VII-deficient recessive DEB keratinocytes of immunodeficient mice [124]. However, in order to avoid the risk of autoimmunity, human trials excluded patients without positive expression to the NC1 domain of collagen VII, the most antigenic portion of this protein [6]. While NC1 is a highly targeted epitope in EBA, only approximately 30% of patient cell cultures fail to express NC1 [125]. Another important criterion was selection of patients with severe generalized recessive DEB showing the absence of expression of full-length collagen VII (near the NC2 domain) [126]. All patients tolerated grafting with collagen-VII-engineered autologous epidermal sheets without adverse events and skin biopsy demonstrated linear collagen VII expression [126,127]. Clinical improvement in wound healing was more profound in grafted sites and patient-reported pain, itch, and wound durability [126]. Only one patient in this study developed autoantibodies (specific to NC2 domain of collagen VII), but pre-therapy serum Western blot analysis showed low levels of transient collagen VII antibodies despite an initial negative screening with indirect immunofluorescence [126]. It is likely that immunoreactivity pre-existed in this patient, and gene therapy exacerbated the immune response. Screening data revealed that this patient expressed a collagen VII molecule containing NC1 domain but not NC2, suggesting that this patient’s reaction was possibly an allo-reaction to the therapeutic gene product [126]. Thus, this patient did not experience increased blistering outside of the treated areas. Although low quantities of collagen VII antibodies in recessive DEB patients are considered nonpathogenic, as in this patient’s case [127,128,129], caution is advised when these patients are treated with gene therapy.

Nonsense mutations, prevalent in 30% of recessive DEB patients, result in a premature termination codon that generates a truncated collagen VII product. In a recent trial of these recessive DEB patients treated with intravenous gentamicin, a premature termination codon readthrough was induced which created a new type VII collagen and anchoring fibrils that persisted for 3 months. Preliminary results demonstrate that none of the patients developed autoantibodies to collagen VII despite aminoglycoside-induced production of new collagen VII [130].

## 9. Conclusions

Although novel treatments appear to be a safe and effective option for select patients, preventing immunoreactivity remains a challenge. The principal goal of gene therapy is to repair or replenish functional protein; however, avoiding highly immunogenic areas of epitope binding is helpful. Future improvements in gene therapy, such as in vivo approaches and refined repair of highly immunogenic areas (NC1 in collagen VII) may help prevent autoimmunity and result in successful gene therapy. Monitoring patients for autoantibody production and regular direct immunofluorescence screening in patients with EB subtypes who present with worsening of skin fragility may be warranted. Such cases may reveal additional examples of EB and AIBD overlap, providing a better understanding of these autoimmune and autoinflammatory responses.

## Figures and Tables

**Figure 1 antibodies-10-00019-f001:**
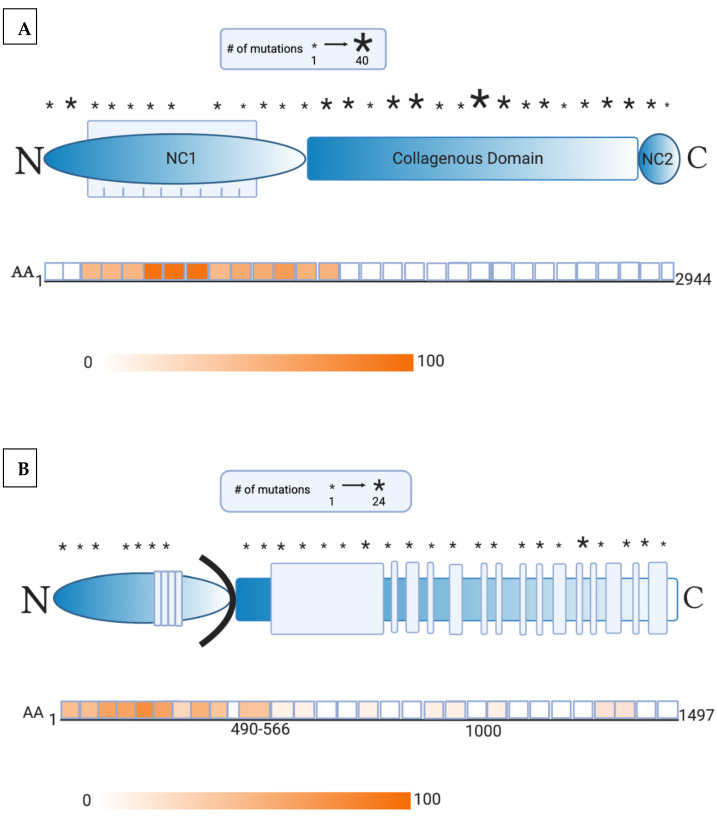
(**A**) Collagen mutation profile and immunogenic domains of epitope reactivity. This schematic representation of collagen VII consists of noncollagenous-1 (NC1, shown as a blue elliptical bar), triple-helix collagenous (shown as a blue rectangular bar), and noncollagenous-2 domains (NC2, shown as a blue oval bar). A crucial region within NC1 is the fibronectin-III-like domains 1–9 (shown as blue vertical bars). The asterisks indicate the approximate location of the mutations alongside intervals of 100 amino acids within the collagen VII polypeptide chain. The size of the asterisk corresponds to the number of mutations detected within the specified interval. The second half of this image depicts the combined results of studies measuring the reactivity of sera from patients with epidermolysis bullosa acquisita to epitopes alongside collagen VII. The intensity of the color relates to the percent reactivity identified within the individual domains. The areas of highest immunogenicity in a majority of patients include fibronectin-III-like domains 4–6 of collagen VII (approximately AA 500–800), and various regions within the NC1 and collagenous domains. As such, gene therapy must target the greatest number of mutations, while avoiding highly immunogenic areas of epitope binding [6,7,8]. (**B**) In this schematic representation of collagen XVII, the extracellular domain consists of stretches of noncollagenous domains and a series of 15 collagenous domains (shown as blue vertical bars). Also represented are the intracellular (shown as a blue elliptical bar) and transmembrane domains (shown as a black curved bar). The asterisks indicate the approximate location of the mutations alongside intervals of 50 amino acids within the collagen XVII polypeptide chain. The size of the asterisk corresponds to the number of mutations detected within the specified interval. The second half of this image depicts the combined results of studies measuring the reactivity of sera from patients with bullous pemphigoid to epitopes alongside collagen XVII. The intensity of the color relates to the percent reactivity identified within the individual domains. The NC16a region of collagen XVII (AA 490–566) was identified as having the highest immunogenicity in the majority of patients but reactivity to various subdomains within the intracellular region must be considered. As such, gene therapy must target the greatest number of mutations, while avoiding highly immunogenic areas of epitope binding [9,10,11,12,13].

**Table 1 antibodies-10-00019-t001:** Select autoantigens shared between pemphigoid diseases and epidermolysis bullosa.

Antigen	EB Subtype	AIBD Subtype
BP230 (dystonin)	EBS	Bullous pemphigoid
Collagen XVII (BP180)	JEB	Bullous pemphigoid, Pemphigoid gestationis
Laminin 332	JEB	Mucous membrane pemphigoid
α6β4 integrin	JEB	Mucous membrane pemphigoid
Collagen VII	DEB	Epidermolysis bullosa acquisita

AIBD = autoimmune blistering disease; EBS = epidermolysis bullosa simplex; JEB = junctional epidermolysis bullosa; and DEB = dystrophic epidermolysis bullosa.

**Table 2 antibodies-10-00019-t002:** Presence of autoantibodies in patients with EB.

	EB Subtype	*n*	Autoantigen												
			Collagen							FN	LAM	Dsg1	Dsg3	Collagen XVII/ BP180	BP230
			I	II	III	IV	V	VI	VII						
**[14]**	EBA	2	0.0%	0.0%	100.0%	0.0%	100.0%	0.0%		0.0%	50.0%				
	EBS	20	0.0%	0.0%	85.0%	60.0%	85.0%	0.0%		0.0%	40.0%				
	JEB	4	0.0%	0.0%	100.0%	50.0%	100.0%	25.0%		0.0%	0.0%				
	DEB	6	0.0%	16.7%	83.3%	33.3%	100.0%	16.7%		16.7%	66.7%				
	Total	32	0.0%	3.1%	87.5%	50.0%	90.6%	6.3%		3.1%	40.6%				
**[15]**	RDEB	19							4.96 U/mL			5.62 U/mL	6.14 U/mL	14.2 U/mL	12.7 U/mL
	Other EB	23							1.08 U/mL			2.67 U/mL	2.8 U/mL	5.7 U/mL	3.7 U/mL
	Healthy Controls	38							0.26 U/mL			2.12 U/mL	1.58 U/mL	1.82 U/mL	1.68 U/mL
**[16]**	RDEB	17							88%					combined percentage of 88%	
	EBS	10							10%					combined percentage of 50%	

Summary of studies assessing the seropositive (%) or quantity (U/mL) of autoantibodies against skin antigens. Abbreviations: Dsg 1 = desmoglein 1; Dsg 3 = desmoglein 3; EBA = epidermolysis bullosa acquisita; EBS = epidermolysis bullosa simplex; FN = fibronectin; JEB = junctional epidermolysis bullosa; and LAM = laminin; RDEB = recessive dystrophic epidermolysis bullosa.

**Table 3 antibodies-10-00019-t003:** Reported cases of confirmed cases of AIBD arising in patients with EB.

Year	Author	EB Type	AIBD Type	Workup
2016	Hayashi	DDEB	EBA	DIF: Linear deposits of IgG and C3 at the DEJ IIF: Linear deposition of IgG at the dermal side of the DEJ Immunoblot analysis: Reactive to collagen type VII and its NC1 domain. Non-reactive to laminin 322 Mutations: c.7868G > A in the COL7A1 gene
2018	Guerra	RDEB	EBA	DIF: Linear deposition of IgG with a u-serrated pattern along the cutaneous BMZ IIF: IgG binding to the dermal side of the salt-split skin ELISA: Positive for anti-collagen type VII, anti-BP180, and anti-BP230 Immunoblot Analysis: Reactive to laminin 332 Mutations: c.410G > A and c.3674C > T in the COL7A1 gene
2019	Fania	JEB	BP	DIF: Linear IgG and C3 deposits in an n-serrated pattern at the DEJ IIF: Epidermal staining of the salt-split skin ELISA: Positive for anti-BP180. Negative for anti-BP230 Immunoblot analysis: Reactive to BP180 and its LAD-1 domain. Not reactive to laminin 332 Mutations: c.1132 + 5G > A in the LAMB3 gene

AIBD = autoimmune blistering disease; DDEB = dominant dystrophic epidermolysis bullosa; RDEB = recessive dystrophic epidermolysis bullosa; JEB = junctional epidermolysis bullosa; EBA = epidermolysis bullosa acquisita; BP = bullous pemphigoid; DIF = direct immunofluorescence; IIF = indirect immunofluorescence; DEJ = dermal epidermal junction; BMZ = basement membrane zone; and EB = epidermolysis bullosa.

## Data Availability

No datasets were generated during this study.

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
