# Peer review of "A Review of Acquired Autoimmune Blistering Diseases in Inherited Epidermolysis Bullosa: Implications for the Future of Gene Therapy"

_2073-4468, 2021, doi:10.3390/antib10020019_

Round 1

Reviewer 1 Report

In general, this is a comprehensive review of a clinically very relevant topic. The authors should adhere to the newest nomenclature of EB according to Has et al. Br J Dermatol 2020.

I have a few minor comments:

Table 1: According to the way the authors denominate proteins Collagen 17 should read Collagen XVII.

Table 2: In the autoantigen row: It should be stated that collagen XVII and BP180 are two names for one protein.

  1. Proinflammatory mechanisms: MMP9 and Il-8 are upregulated not only in the circulation, but also in situ in the blister fluid of EBS, JEB and DEB.

Please cite:

MMP-9 and CXCL8/IL-8 are potential therapeutic targets in epidermolysis bullosa simplex. Lettner T et al. PLoS One. 2013 Jul 19;8(7):e70123

Increased levels of matrix metalloproteinase-9 and interleukin-8 in blister fluids of dystrophic and junctional epidermolysis bullosa patients. Lettner T et al J Eur Acad Dermatol Venereol. 2015 Feb;29(2):396-398.

Page 8, line 243: Antibiotic treatment in type XVII collagen JEB: In the reference 60 I could not find such an experiment. It is rather that this reference refers to Goldstein AM, Davenport T, Sheridan RL (1998) Junctional epidermolysis bullosa:Diagnosis and management of a patient with the Herlitz variant.J Pediatr Surg33:756–758. This is a single patient report and should not form the base of the assumption that antibiotic treatment improves JEB by an antibacterial effect.

Figure 1:

….gene therapy must target ……avoiding highly immunogenic areas of epitope binding.

Anticipating the induction of autoantibodies to a replenished protein in gene therapy this statement is correct. However, this is not the main consideration in gene therapy. The first consideration is always to have a functional protein. A protein without immunogenic epitopes is nice to have, but potentially useless in terms of function.

References:

I think 110 is out of place.

Author Response

Response to Reviewer 1 Comments

Thank you so much for your feedback and helping us improve this manuscript.

Point 1: In general, this is a comprehensive review of a clinically very relevant topic. The authors should adhere to the newest nomenclature ofEB according to Has et al. Br J Dermatol 2020.

Response 1: Thank you for suggesting this article. We have edited EB nomenclature throughout our manuscript to only include EBS, JEB, and DEB. Pg 7 Line 359, we changed GABEB to JEB. Pg 8, Line 521, we changed EBS-DM to EBS.

Point 2: I have a few minor comments:Table 1: According to the way the authors denominate proteins Collagen 17 should read Collagen XVII.

Response 2: Thank you for the observation. We have edited it to collagen XVII, pg.2 line 68.

Point 3: Table 2: In the autoantigen row: It should be stated that collagen XVII and BP180 are two names for one protein.

Response 3: Thank you for the observation. We have added collagen XVII/BP180 to Table 2, pg 4.

Point 4: Proinflammatory mechanisms: MMP9 and Il-8 are upregulated not only in the circulation, but also in situ in the blister fluid of EBS, JEB and DEB.

Please cite:MMP-9 and CXCL8/IL-8 are potential therapeutic targets in epidermolysis bullosa simplex. Lettner T et al. PLoS One. 2013 Jul 19;8(7):e70123

Increased levels of matrix metalloproteinase-9 and interleukin-8 in blister fluids of dystrophic and junctional epidermolysis bullosa patients. Lettner T et al J Eur Acad Dermatol Venereol. 2015 Feb;29(2):396-398.

Response 4: Thank you for adding that important point to the manuscript. We have added citations and changed the paragraph to include MMP-9 and IL-8 to section 5pg 8,lines 502-6.

“Similarly, MMP-9 and IL-8 are not only upregulated in circulation, but also in the blister fluid of EBS, JEB and DEBpatients[85,86]. Both MMP-9 and IL-8 have been reported in other AIBD, such as BP and pemphigus vulgaris but their pathologic significance in EB and overlapping AIBD needselucidation [68,87,88].”

Point 5: Page 8, line 243: Antibiotic treatment in type XVII collagen JEB: In the reference 60 I could not find such an experiment. It is rather that this reference refers to Goldstein AM, Davenport T, Sheridan RL (1998) Junctional epidermolysis bullosa:Diagnosis and management of a patient with the Herlitz variant.J Pediatr Surg33:756758. This is a single patient report and should not form the base of the assumption that antibiotic treatment improves JEB by an antibacterial effect.

Response 5: Thank you for pointing out our error. We have editedpg 9,line 677-9and removed the case report from consideration.

“Further strengthening the importance of alteration of microbial diversity in BP is a study that reports clinical improvement of BP with antibiotics [118].”

Point 6: Figure 1:....genetherapy must target ......avoiding highly immunogenic areas of epitope binding.

Anticipating the induction of autoantibodies to a replenished protein in gene therapy this statement is correct. However, this is not the main consideration in gene therapy. The first consideration is always to have a functional protein. A protein without immunogenic epitopes is nice to have, but potentially useless in terms of function.

Response 6: We agree with your aforementioned statement and have rephrased the sentence as follows inpg 10lines 848-964.

“The principal goal of gene therapy is to repair or replenish functional protein, however avoiding highly immunogenic areas of epitope binding is helpful.”

Point 7:References:

I think 110 is out of place.

Response 7:Thank you for your observation. We have removed 110 from the article and replaced it with the correct citation inpg 10 line 823.“Another important criterion was selection of patients with severe generalized recessive DEB showing the absence of expressionof full-length collagen VII (near the NC2 domain) were selected [126]

Reviewer 2 Report

The authors present an interesting review of acquired autoimmune disease in inherited EB and relate the findings to gene therapy approaches. This is a well written review and it was a pleasure to read. It was great to see more attention to the autoimmunity in skin blistering diseases. The tables are well presented, there are only minor suggestions which I think could improve the flow of the review that are required:

- Section describing the mechanism of autoantibody mediated tissue damage in skin blistering diseases would be useful as well as a schematic (see https://doi.org/10.3389/fimmu.2019.01089 and Figure 1).

 - Figure 1 and 2 should be combined and brought in earlier in the manuscript. 

- Authors should mention studies surrounding the role of Flightless protein in skin blistering showing first mechanism of resolution of blistering and inflammation in EBA. Flightless has also been shown to be elevated in human inherited EB especially DEB and has a role in autoimmunity in AD which may explain some observations in skin blistering diseases. 

- Last paragraph should be separated as a conclusion subheading and further expanded to what authors feel is next direction in the field. At the moment article has an abrupt finish with no overall direction of what is lacking in the field or what next research direction should be.

- Excellent review overall, well done. 

Author Response

Point 1: The authors present an interesting review of acquired autoimmune disease in inherited EB and relate the findings to gene therapy approaches. This is a well written review and it was a pleasure to read. It was great to see more attention to the autoimmunity in skin blistering diseases. The tables are well presented, there are only minor suggestions which I think could improve the flow of the review that are required:

Response 1: We thank the author for your thoughtful feedback and helping us add more valuable information to our discussion.

Point 2: - Section describing the mechanism of autoantibody mediated tissue damage in skin blistering diseases would be useful as well as a schematic (see https://doi.org/10.3389/fimmu.2019.01089 and Figure 1).

Response 2: Considering your important point, we have included a section on autoantibody tissue damage in EBA in pg5, lines 167-174. For BP, we mentioned this point in lines 175-182 and within several sections of the manuscript. We believe that a schematic will be out of the scope of this paper and have cited detailed reviews describing these mechanisms in detail.

“In epidermolysis bullosa acquisita (EBA), autoantibody-induced tissue damage against collagen VII contributes to blistering [22]. Complement activation through both the classical and alternative pathways have been implicated in disease pathogenesis but the alternative pathway appears to be predominant [23,24]. Autoreactive IgG and immune complex-FcγR binding initiate an inflammatory complement cascade resulting in extravasation of neutrophils, release of proteolytic enzymes, and reactive oxygen species [25]. Moreover, T cells may perpetuate tissue damage in EBA through association with immune complexes and neutrophils [26]. “

Point 3: Figure 1 and 2 should be combined and brought in earlier in the manuscript. 

Response 3: Thank you for your wonderful suggestion. We have combined the two figures and moved them at the end of section 1, lines 57-58.

“For illustration, Figure 1 details the location of mutations in collagen VII and collagen XVII with respect to the immunogenic epitopes in AIBD”

Point 4: Authors should mention studies surrounding the role of Flightless protein in skin blistering showing first mechanism of resolution of blistering and inflammation in EBA. Flightless has also been shown to be elevated in human inherited EB especially DEB and has a role in autoimmunity in AD which may explain some observations in skin blistering diseases.

Response 4: Thank you for mentioning this important point. We really enjoyed reading more about this protein. We added it to Section 3, pg 7, Lines 310-320 as seen below.

“Flightless I (Flii), an actin remodeling protein and nuclear receptor co‐activator has been implicated in cell adhesion and intracellular signaling [55,56]. Disruptions in levels of Flii affect wound healing through effects on TGFβ signaling and ECM reorganization [56]. As such, levels of Flii are increased in blistered skin of EB patients, and Flii over‐expression in experimental EBA decreased expression of proteins that make up cell–cell tight junctions [57]. The interaction of Flii with tight junctions may impair formation of tight junction protein complexes and function of the epidermal barrier [57]. Moreover, topical application of Flii neutralizing antibodies improved healing in experimental EBA [56]. Additionally, increased levels of Flii have been shown to result in increased pro-inflammatory cytokine production in a Th2 pattern in atopic dermatitis, and poor mucosal healing in ulcerative colitis [58,59]. Thus, altered levels of Flii in EB and EBA may contribute to the delayed healing and skin fragility [57].”

Point 5: - Last paragraph should be separated as a conclusion subheading and further expanded to what authors feel is next direction in the field. At the moment article has an abrupt finish with no overall direction of what is lacking in the field or what next research direction should be.

Response 5: Yes, we agree with you and have added more points to the conclusion to allow the paper to have a smoother transition in pg 10, lines 817-941.

“Although novel treatments appear to be a safe and effective option for select patients, preventing immunoreactivity remains a challenge. The principal goal of gene therapy is to repair or replenish functional protein; however, avoiding highly immunogenic areas of epitope binding is helpful. Future improvements in gene therapy, such as in vivo approaches and refined repair of highly immunogenic areas (NC1 in Collagen VII) may help prevent autoimmunity and result in successful gene therapy. Monitoring patients for autoantibody production and regular direct immunofluorescence screening in patients with EB subtypes who present with worsening of skin fragility may be warranted. Such cases may reveal additional examples of EB and AIBD overlap, providing a better understanding of these autoimmune/autoinflammatory responses.”
